# Optimized Selective Media Enhance the Isolation and Characterization of Gut-Derived Probiotic Yeasts

**DOI:** 10.3390/jof11120885

**Published:** 2025-12-15

**Authors:** Kevin Mok, Kwantida Popitool, Areerat Songla, Tawisa Pongsuwanporn, Pitchsupang Torrungruang, Sunee Nitisinprasert, Jiro Nakayama, Massalin Nakphaichit

**Affiliations:** 1Department of Biotechnology, Faculty of Agro-Industry, Kasetsart University, Bangkok 10900, Thailand; kevin.m@ku.th (K.M.); kwantida109@gmail.com (K.P.); areerat.songla10@gmail.com (A.S.); tawisa.po@ku.th (T.P.); pitchsupang.t@ku.th (P.T.); fagisnn@ku.ac.th (S.N.); 2Center of Excellence for Microbiota Innovation, Faculty of Agro-Industry, Kasetsart University, Bangkok 10900, Thailand; 3Department of Bioscience and Biotechnology, Faculty of Agriculture, Kyushu University, Fukuoka 819-0395, Japan; nakayama@agr.kyushu-u.ac.jp

**Keywords:** yeast, gut fungi, mycobiome, probiotic yeast

## Abstract

This study applied a guided culturomics workflow to isolate and characterize gut-associated yeasts as probiotic candidates. Culture conditions were optimized using Dixon agar and Modified Schädler Agar with a moderate antibiotic cocktail (colistin, chloramphenicol, vancomycin) to suppress bacteria without impairing yeast growth, combined with incubation at 37 °C under anaerobic conditions to mimic the intestinal environment. From three healthy donors, 305 isolates were recovered (MSA: 193; Dixon: 112). After excluding pseudohyphal morphotypes and PCR positive *Candida* isolates, 127 non-*Candida* strains remained. Safety screening (hemolysis, DNase, coagulase) reduced the pool to 26 safe isolates. Simulated gastrointestinal stress tests showed that 20 out of 26 exhibited at least 50 percent survival under pH 2.0 or 0.5% bile salts. Functional assays revealed strain-specific antimicrobial activity against *Staphylococcus aureus*, *Escherichia coli* including O157:H7 and *Salmonella* species, with several isolates (Y6, Y22, Y42, Y48, Y55, Y56, Y73, Y105, Y127) showing broad and strong inhibition. Two isolates Y44 and Y55 further demonstrated robust bile and acid tolerance (>50% survivability) in both conditions. All isolates displayed intracellular but not extracellular bile salt hydrolase activity, indicating a viability dependent cholesterol lowering potential. Overall, the workflow minimized bacterial interference while enriching for safe and functional yeasts.

## 1. Introduction

The human gut microbiota has long been recognized for its profound role in maintaining health and influencing disease states, primarily focusing on bacterial communities. However, recent advancements in microbial ecology, particularly through culture-independent methods like metagenomics, have broadened our understanding, revealing that the gut microbiota is far more diverse than previously known [1,2]. Among the findings is the identification of yeasts as significant constituents of the gut ecosystem, previously underrepresented in culture-dependent studies [3,4]. This insight challenges the conventional microbiota paradigm, which traditionally emphasizes bacteria as the dominant players in gut health and opens new avenues for research on non-bacterial gut residents, such as fungi, particularly yeasts, as potential probiotics.

Traditionally, studies of the human gut microbiota relied heavily on culture-based techniques, which provided a limited view of microbial diversity, as many organisms in the gut are fastidious or non-culturable. The advance of next-generation sequencing (NGS) and metagenomic analysis has revolutionized this field by enabling the comprehensive profiling of microbial communities without the need for culturing. Previous study has demonstrated that culture-independent techniques reveal a much more complex gut mycobiome, including previously unidentified fungal species, with *Saccharomyces*, *Candida*, and *Debaryomyces* genera prominently featured [5]. These findings underscore the inadequacy of traditional culturing methods, which often failed to detect these microorganisms. The use of metagenomic data has now paved the way for a more inclusive understanding of the human gut microbiota, showing that yeasts, alongside bacteria, play important roles in maintaining gut homeostasis and possibly contributing to metabolic health [6].

Moreover, several studies have further highlighted that gut-associated yeasts are not merely transient dietary contaminants but may form stable populations within the gut microbiome [7,8]. These species interact with bacterial communities, influencing metabolic pathways, immune responses, and even gut barrier functions. Yeasts like *Candida albicans* and *Saccharomyces boulardii* have been particularly well studied, the latter already recognized as a probiotic with clinical applications in the prevention and treatment of gastrointestinal disorders [9]. Yet, many other yeast species in the gut remain largely unexplored, representing a vast, untapped reservoir of potential health-modulating organisms.

The emerging field of culturomics, which aims to complement metagenomic data by improving microbial cultivation techniques, has begun to address the gaps in our ability to culture diverse microorganisms, including yeasts. Culturomics employs high-throughput culturing strategies and customized growth media to cultivate previously unculturable species, thereby enhancing the scope of microbial isolation and characterization. Through this approach, previous study successfully cultured over 30 novel bacterial species from the human gut [10,11]. This success in culturing bacterial species suggests that similar strategies can be applied to the isolation of yeasts, with the potential to discover new fungal strains that may hold probiotic properties.

Our study aims to extend the application of culturomics to the targeted isolation of gut-associated yeasts. By leveraging data from previous metagenomic studies, we have designed a specialized condition tailored to support the growth of yeasts in the gut environment [12]. This condition consists of medium formulation, and environmental factors are based on optimizing key nutrients and growth factors known to support fungal proliferation while mimicking gut conditions. Through this approach, we anticipate the isolation of novel yeast strains that may serve as future probiotic candidates. This work is particularly significant in the context of the growing recognition of fungi’s role in gut health and their potential therapeutic applications.

Probiotics have predominantly been derived from bacterial strains, such as *Lactobacillus* and *Bifidobacterium*. However, yeast species like *Saccharomyces boulardii* have proven that fungi can also be effective probiotics, with demonstrated benefits in preventing diarrhea, regulating the immune system, and modulating gut microbiota composition [13]. The expanding knowledge of gut-associated yeasts offers new opportunities to explore other fungal strains for probiotic use. For instance, *Debaryomyces hansenii*, a yeast frequently found in fermented foods and detected in human gut microbiota, may have unrevealed probiotic potential due to its ability to tolerate and metabolize bile salts, a feature important for gut colonization and survival [14].

Yeast-based probiotics offer distinct advantages over bacterial probiotics, as yeasts are naturally more resistant to environmental stresses such as pH fluctuations and bile salts, enhancing their survival in the gastrointestinal tract [15,16]. Certain yeast species can also modulate gut microbiota by producing bioactive compounds like organic acids and vitamins, which promote microbial diversity and metabolic function [17]. Additionally, yeasts do not harbor antibiotic resistance genes, a critical safety concern for bacterial probiotics, making them a safer alternative, particularly for immuno-compromised individuals [18,19].

Given the growing evidence that yeasts are an integral part of the gut microbiome, their underrepresentation in gut microbiota research warrants further investigation. This study aims to address this gap by utilizing advanced culturomics techniques to isolate and characterize gut-associated yeasts, ultimately contributing to the development of novel yeast-based probiotics. The potential health benefits of yeast in gut homeostasis, immune regulation, and microbial modulation could lead to the next generation of probiotics that move beyond bacteria, offering more robust and targeted interventions for gastrointestinal health.

## 2. Materials and Methods

### 2.1. Antibiotic Susceptibility Testing and Growth Assessment of Yeast Isolates

In this study, three antibiotics, namely chloramphenicol (Sigma-Aldrich, St. Louis, MO, USA), vancomycin (Goldbio, St. Louis, MO, USA), and colistin (Goldbio, St. Louis, MO, USA), were tested for their ability to suppress bacterial growth while minimizing their impact on yeast. Each antibiotic was applied at three concentrations, chosen according to the Clinical & Laboratory Standards Institute (CLSI) susceptibility breakpoint guidelines for *Enterobacteriaceae* [20,21] (Table 1). The *Enterobacteriaceae* is a bacterial group that is abundantly represented in the human gut and frequently overgrows on non-selective media during yeast isolation. Additionally, the combination of colistin, chloramphenicol, and vancomycin was strategically selected because it effectively suppresses a broad range of dominant gut bacteria, including Gram-negative and Gram-positive bacteria.

Antibiotic susceptibility of yeast isolates was initially assessed by agar well diffusion. Four yeast commonly associated with the human gut (*Saccharomyces cerevisiae* YC1, *Candida albicans* PCJ94-2, *Candida tropicalis* PYJ100-2, and *Rhodotorula mucilaginosa* TBRC-4420) were cultured in Yeast Extract Peptone Dextrose (YPD) broth (Difco Laboratories, Detroit, MI, USA) at 30 °C for 48 h. *Saccharomyces cerevisiae* YC1 was obtained from the Culture Collection of the Center of Excellence for Microbiota Innovation, Kasetsart University, while *Candida albicans* PCJ94-2 and *Candida tropicalis* PYJ100-2 were obtained from the Biodiversity Center, Kasetsart University (BDCKU) [22].

Cells were harvested by centrifugation at 10,000× *g* for 10 min at 4 °C, washed twice with Butterfield’s phosphate buffer, and adjusted to 10^7^ CFU/mL based on OD_420_ nm. Aliquots of 100 µL were spread evenly onto YPD agar plates (Difco Laboratories, Detroit, MI, USA). Wells of 7 mm diameter were prepared using sterile cork borers (Nonaka Rikaki, Tokyo, Japan), and each well was filled with an antibiotic mixture at the designated concentration. Plates were incubated at 30 °C for 48 h, and inhibition was evaluated by measuring the presence of clear zones surrounding the wells. For bacterial controls, *Escherichia coli* ATCC 8739, *Salmonella enteritidis* S003, and *Staphylococcus aureus* ATCC 6538p were grown on Mueller Hinton Agar (Oxoid, Basingstoke, UK) at 37 °C for 24 h under the same conditions.

Antibiotic concentrations that able to inhibit bacteria growth but did not produce inhibition zones against yeast isolates were further evaluated for their effect on yeast growth kinetics. Yeast strains were precultured in YPD broth at 30 °C for 48 h and subsequently inoculated into fresh YPD broth supplemented with the selected antibiotic concentrations, adjusted to a final inoculum of 10^4^ CFU/mL. Cultures were incubated at 30 °C, and aliquots were collected at 0, 9, 18, 24, and 48 h. Each aliquot was serially diluted and spread onto YPD agar plates to determine viable counts expressed as CFU/mL. Yeast grown in antibiotic-free YPD broth served as the control. Growth curves were constructed to assess whether the selected antibiotic concentrations influenced yeast proliferation.

### 2.2. Fecal Suspension and Preparation

Fecal samples were collected from three healthy volunteers who met the following criteria: BMI between 18.5 and 25, no consumption of probiotics or prebiotics for at least one month, no antibiotic use for at least three months prior to the study, and no history of gastrointestinal or non-communicable diseases. The research protocol, including the collection of fecal samples from the volunteers, was approved by the Ethics Committee of Kasetsart University (License number COA64/068) on 8 December 2021; Clinical trial number: not applicable. Written informed consent was obtained from all participants before sample collection. Each volunteer was provided with a fecal collection tube and flushing paper and instructed to collect the sample as soon as possible. The samples were stored in an anaerobic jar on ice until processing in the laboratory.

Sample preparation was done according to previously described protocol [23]. In brief, 5 g of feces were suspended in 20 mL of Butterfield’s phosphate buffers and homogenized using a stomacher (Stomacher 400; Seward, Worthing, UK) at maximum speed and filtered by stomacher bag (Interscience, Saint Nom la Bretêche, France). The liquid phase, obtained by filtering out solid fecal residues, was used for yeast enumeration in further optimization of growth conditions and isolation with the optimized medium.

### 2.3. Optimization of Growth Condition Based on Media, Temperature, Antibiotic Dose, and Incubation Atmosphere

The media used in this study were based on a previous study that incorporated high throughput and culturomics in human gut bacteria and fungi study [3,24,25]. We evaluated the suitability of Dixon agar and Modified Schadler Agar (MSA), as these media previously shown to support the growth of diverse fungal genera. Both media facilitate the growth of fastidious, protein and lipid-dependent yeasts commonly found in the human gut. Dixon agar provides lipid-rich components and complex nitrogen sources, whereas MSA offers balanced peptones, glucose, and essential vitamins, allowing broader recovery of diverse fungal taxa. These characteristics make both media highly suitable for culturomics-based approaches targeting gut-associated yeasts. Additionally, two antibiotic concentration ranges (low and moderate) were selected for evaluation, following the preliminary screening results from the antibiotic susceptibility testing and growth assessment, together with two incubation temperatures (30 °C and 37 °C), representing optimal conditions for yeast growth and human body temperature, respectively. Fecal suspensions from three donors were plated according to the conditions outlined in Table 1. After 48 h of incubation, the number of yeast colonies grown under each condition was calculated.

Optimization of yeast growth conditions was performed by evaluating medium composition, antibiotic concentration, and incubation temperature. Two media, Dixon agar and Modified Schadler Agar (MSA), previously reported to support diverse fungal genera [24], were selected. Each medium was supplemented with two antibiotic concentration ranges (low and moderate) and incubated at two temperatures (30 °C and 37 °C), representing optimal yeast growth and human body temperature, respectively. Fecal suspensions obtained from three donors were plated under these conditions (Table 1). After 48 h of incubation, yeast colony counts were recorded for each treatment, and results were compared to identifying the most suitable growth condition.

To further evaluate growth differences under oxygen availability, yeast cultures were examined using the optimized condition identified. For aerobic growth, plates were incubated under atmospheric conditions. For anaerobic growth, plates were incubated in an anaerobic chamber containing a gas mixture of 90% N_2_, 5% H_2_, and 5% CO_2_ at controlled concentrations. All plates were prepared using the most appropriate medium, antibiotic dose, and incubation temperature as determined in the initial optimization.

### 2.4. Isolation Gut Yeast from Human Fecal Samples

The optimized conditions were then used to isolate yeast from the human gut. Briefly, fecal samples from three volunteers were processed according to the method outlined in Section 3. After a 48 h incubation, yeast isolates were selected based on their morphological characteristics, such as colony color, size, opacity, and surface consistency. The isolates were re-streaked onto half-strength YPD agar and incubated at 37 °C for 72 h to promote morphological differentiation. After incubation, colonies were examined microscopically. Yeasts that did not produce pseudohyphae or hyphae were selected for further characterization. Given that pseudohyphal formation may reflect increased adhesive or invasive potential, these isolates were excluded as a precaution to maintain a conservative safety threshold for probiotic selection

### 2.5. Identification of Gut Yeast from Human Fecal Samples

A loopful of each yeast isolate was inoculated into YPD broth and incubated for 24 h. One milliliter of culture was harvested by centrifugation (12,000× *g*, 5 min), and the pellet was washed twice with Butterfield’s phosphate buffer. Cells were resuspended in ASL lysis buffer (Qiagen, Hilden, Germany) and transferred to 2 mL screw-cap tubes containing sterile zirconia beads (0.3 mm and 0.7 mm, 0.3 g each; BioSpec, Bartlesville, OK, USA). Cell disruption was performed using a FastPrep-24 homogenizer (MP Biomedicals, Santa Ana, CA, USA) at 6.5 m/s for 5 min, with one-minute beating cycles followed by five minutes on ice. Lysates were centrifuged (16,000× *g*), and the supernatant was treated with 10 M ammonium acetate prior to a second centrifugation. DNA was precipitated with an equal volume of isopropanol, resuspended in 50 µL TE buffer, and used as template DNA.

PCR amplification was carried out using 2× Taq Plus Master Mix (Vazyme, Nanjing, China) with primer pairs SI4F/SI7R (for *Saccharomyces* sensu stricto) and 5.8S-1F/28S-1R (for *Candida* spp.), following published protocols [26,27]. *Saccharomyces* sp. YC1 and *Candida albicans* PCJ-942 served as the positive control, and a no-template control (NTC) was included. Amplicons were resolved on 2.5% agarose gels at 90 V for 40 min. Clear single bands of ~200 bp (*Saccharomyces* sensu stricto) or ~150 bp (*Candida* spp.) were considered positive, while absence of amplification in both groups was designated as “other” yeasts.

### 2.6. Probiotic Characterization

#### 2.6.1. Assessment of Safety Profiles in Yeast Isolates

The safety of yeast isolates was evaluated through hemolytic and DNase activity assay [28]. Each isolate was first cultured in YPD broth for 24 h and subsequently streaked onto the respective media.

For hemolytic activity, isolates were streaked onto Columbia blood agar plates (Difco Laboratories, Detroit, MI, USA) supplemented with 5% sheep blood and incubated anaerobically at 37 °C for 48 h. *Staphylococcus aureus* ATCC 6538p and *Lactiplantibacillus rhamnosus* ATCC 53103 served as positive and negative controls, respectively. Isolates producing a clear hemolytic zone were classified as hemolytic and excluded from further analysis.

DNase activity was assessed by inoculating isolates onto DNase test agar (Difco Laboratories, Detroit, MI, USA) containing methyl green, followed by anaerobic incubation at 37 °C for 48 h. The presence of a clear zone surrounding colonies indicated DNase activity and potential virulence.

#### 2.6.2. Coagulase Activity

Coagulase activity was included as an additional precautionary measure to screen for any unexpected virulence-associated traits. The coagulase activity was determined using rabbit coagulase plasma with EDTA (Difco Laboratories, Detroit, MI, USA). For the slide test, a drop of plasma was mixed with a colony, and visible clumping within 1 min indicated a positive reaction. Negative results were further examined by adding 100 µL of overnight broth culture to 0.5 mL of reconstituted rabbit coagulase plasma in a test tube and monitoring at 0, 4, and 24 h. Isolates showing no coagulation were considered safe and selected for subsequent experiments [29,30].

#### 2.6.3. Acid and Bile Salt Tolerance Assay of Yeast Isolates

The assessment was conducted according to previously described methods with modification [31]. Briefly, yeast isolates were cultured by inoculating a single colony into 10 mL of YPD broth and incubating at 30 °C for 24 h. Subsequently, 2% (*v*/*v*) of the culture was transferred into fresh YPD broth and incubated under the same conditions for an additional 24 h to standardize cell growth. Following incubation, 0.5 mL of the culture was centrifuged at 6000× *g* for 5 min, and the supernatant was discarded.

The resulting pellet was washed twice with phosphate-buffered Butterfield’s buffer (PB buffer) to remove residual medium. The cells were then resuspended in 5 mL of one of the following media: YPD adjusted to pH 2.0 using 5 M hydrochloric acid (HCl), YPD supplemented with 0.5% (*w*/*v*) bile salts (Sigma-Aldrich, St. Louis, MO, USA), or standard YPD (pH 6.5) as the control. Cultures were incubated anaerobically at 37 °C for 24 h.

Cell survival was determined by measuring optical density at 600 nm (OD600). The survival rate was calculated using the following equation:$$\%\;\mathrm{Survival}\;=\frac{N2}{N1}\times 100$$
where N2 represents the OD value of isolates grown in YPD supplemented with bile salts or adjusted to pH 2.0, and N1 represents the OD value of isolates grown in standard YPD.

#### 2.6.4. Assessment of Anti-Pathogenic Activity Against Enteropathogenic Bacteria

The anti-pathogenic activity of yeast isolates was assessed against *Staphylococcus aureus* ATCC 6538p, *Escherichia coli* ATCC 8739, *E. coli* O157:H7, *Salmonella enteritidis* S003, and *Salmonella enterica* serovar Typhimurium DMST 48437 using an agar well diffusion assay [32] with slight modifications. Pathogens were cultured in nutrient broth at 37 °C for 18 h, adjusted to OD_600_ = 0.132, and spread onto Mueller–Hinton agar. Wells (7 mm) were filled with 100 µL of yeast culture supernatant (24 h in YPD, pH 6–7). Chloramphenicol (100 µg/mL) and Butterfield’s phosphate buffer served as positive and negative controls, respectively. Plates were incubated at 37 °C for 18 h, and inhibition zones were measured. Yeast isolates showing inhibition were considered to exhibit anti-pathogenic potential.

#### 2.6.5. Assessment of Cholesterol-Lowering Potential of Yeast Based on BSH Activity

Extracellular BSH activity was determined by streaking isolates onto YPD agar supplemented with 0.5% (*w*/*v*) taurodeoxycholic acid (TDCA; Sigma-Aldrich, St. Louis, MO, USA) and 0.01% CaCl_2_, followed by anaerobic incubation at 37 °C for 72 h. Precipitate formation around colonies indicated positive activity, and the diameter of the precipitation zone was recorded.

For intracellular BSH activity, isolates were grown in YPD broth for 24 h, harvested by centrifugation (15,000× *g*, 15 min, 4 °C), washed twice, and resuspended in Butterfield’s phosphate buffer [33]. Cell disruption was achieved with glass beads (0.3 g each of 0.1 mm and 1 mm; BioSpec, Bartlesville, OK, USA) using a FastPrep-24 homogenizer (MP Biomedicals, Santa Ana, CA, USA) at 6.5 m/s for 2 min. Lysates were centrifuged (15,000× *g*, 15 min, 4 °C), and supernatants were stored at −20 °C.

Intracellular enzyme activity was assayed by mixing 0.4 mL of cell-free extract with 0.2 mL of 6 mM TDCA and 1 mL of phosphate buffer (pH 6.0), followed by incubation at 37 °C for 30 min. Reactions were stopped with 0.5 mL of 30% (*w*/*v*) trichloroacetic acid. For quantification, 0.6 mL of the reaction mixture or a taurine standard (2 mg/mL; Sigma-Aldrich) was combined with 1 mL ninhydrin reagent and 1 mL distilled water, boiled for 14 min, cooled, and observed for color change. Development of a purple color indicated positive BSH activity, reflecting release of amino acids from conjugated bile salts.

### 2.7. Statistical Analysis

All statistical analyses were performed using GraphPad Prism version 8.4.3 (GraphPad Software, San Diego, CA, USA). Data are expressed as mean ± standard deviation (SD) unless otherwise specified. Growth curves and inhibition zones were compared using two-way ANOVA followed by post hoc multiple comparison tests.

For experiments involving multiple factors (medium type, temperature, and antibiotic dose), three-way ANOVA was applied to analyze collectively and determine the combination that maximize yeast recovery. Growth under aerobic and anaerobic conditions were analyzed using paired *t*-tests to account for differences in oxygen availability. A *p* value < 0.05 was considered statistically significant.

## 3. Results

### 3.1. Optimization of Selective Medium and Conditions

To maximize the selective recovery of gut-derived yeasts, we systematically examined the influence of antibiotic concentration, culture medium, incubation temperature, and oxygen availability. These factors were chosen to represent critical parameters affecting both bacterial suppression and yeast proliferation in fecal samples. By evaluating their individual and combined effects, the most favorable conditions for isolating yeasts while minimizing bacterial interference were identified.

#### 3.1.1. Antibiotic Concentration Screening

To establish suitable antibiotic concentrations for yeast isolation, bacterial and yeast reference strains were tested against low, moderate, and high doses. All bacterial strains (*Escherichia coli* ATCC 8739, *Salmonella enteritidis* S003, and *Staphylococcus aureus* ATCC 6538P) were inhibited at all concentrations, with inhibition zones increasing in a dose-dependent manner (Table 2). Among them, *S. aureus* exhibited the lowest susceptibility, while *E. coli* and *S. enteritidis* displayed broader inhibition zones. In contrast, most yeast strains (*Candida albicans* PCJ94-2, *Candida tropicalis* PYJ100-2, and *Saccharomyces cerevisiae* YC1) remained unaffected at all doses, whereas *Rhodotorula mucilaginosa* TBRC-4420 was inhibited only at the highest dose. These findings suggest that moderate antibiotic concentration is sufficient to suppress bacterial growth, particularly Gram-negative taxa that are common in the gut, while maintaining yeast viability.

Although pure yeast cultures tolerated the antibiotic at moderate concentration, it is well recognized that antibiotics can reduce yeast growth rates. To assess this effect, representative strains were cultured in YPD broth supplemented with low and moderate antibiotic concentrations and compared to the control (YPD without antibiotic) (Figure 1). The growth profiles of all tested strains were comparable to the control, indicating that neither concentration significantly affected yeast growth or viability. Both antibiotic doses were therefore considered suitable for subsequent experiments on yeast isolation. Nevertheless, given that some gut bacteria may exhibit intrinsic resistance to low antibiotic concentrations, the moderate concentration was selected for further studies.

#### 3.1.2. Influence of Medium, Temperature, and Antibiotic Dose Under Aerobic Condition

The effects of medium type, temperature, and antibiotic dose on bacterial growth (log CFU/mL) are summarized in Table 3. All three factors significantly influenced growth (*p* < 0.05). Among them, antibiotic dose had the greatest impact, explaining 54.15% of the total variation, followed by temperature (16.08%) and medium type (15.21%). This underscores the dominant role of antibiotic concentration in regulating bacterial growth.

Analysis of interaction effects revealed significant interactions between temperature and antibiotic dose (*p* < 0.05), medium type and antibiotic dose (*p* < 0.05), and the three-way interaction of medium type × temperature × antibiotic dose (*p* < 0.05). In contrast, the interaction between temperature and medium type was not significant (*p* > 0.05), suggesting that temperature influences bacterial growth consistently across media. Additionally, volunteers (individual donors) accounted for only 0.1080% of the total variance, indicating that inter-individual differences contributed minimally relative to the pronounced effects of medium type, temperature, and antibiotic dose. Collectively, these results demonstrate that although medium type can influence isolation outcomes, its impact is highly context dependent. Temperature exerts a consistent and measurable effect, whereas antibiotic dose remains the dominant factor governing yeast recovery, with donor variation playing only a minor role. The yeast recovery rates for all medium combinations within the three-factor ANOVA provided in Appendix A.

Multiple comparisons confirmed that yeast growth was significantly enhanced at 37 °C compared to 30 °C across both media (*p* < 0.05), indicating better adaptation to body temperature. Similarly, the moderate antibiotic dose consistently supported higher yeast counts than the low dose, suggesting that insufficient bacterial suppression at low concentrations may reduce yeast recovery. Between the two media, both Modified Schadler Agar (MSA) and Dixon agar supported yeast growth, with differences influenced by antibiotic dose and temperature (Figure 2). Collectively, these results identify 37 °C and moderate antibiotic dose as the optimal conditions for selective yeast isolation from fecal samples.

#### 3.1.3. Effect of Oxygen Availability

Given the predominantly anaerobic nature of the gut, yeast growth was further assessed under aerobic and anaerobic incubation in optimized media. Across both MSA and Dixon, yeast growth (log CFU/mL) was significantly higher under anaerobic compared with aerobic conditions (*p* < 0.05) (Figure 3). This suggests that fecal yeasts possess physiological adaptations for survival in oxygen-limited environments. The combination of anaerobic incubation at 37 °C with moderate antibiotic concentration therefore provides the most favorable condition for recovery of gut-adapted yeast strains.

### 3.2. Isolation and Identification Gut Yeast from Human Fecal Samples

Yeasts were successfully isolated from fecal samples of three donors using optimized conditions on Dixon agar and MSA. A total of 305 isolates were recovered, comprising 193 from MSA and 112 from Dixon, demonstrating that both media can support the growth of gut-associated yeasts. Colony morphology was diverse, and microscopic observation of Gram-stained cells (Figure 4A) revealed that most isolates displayed coccus or ovoid morphology. A subset of isolates exhibited pseudohyphal structures, a trait commonly associated with invasive or pathogenic potential. To ensure biosafety and focus on commensal candidates, 273 isolates without pseudohyphae were retained for further analysis.

Molecular identification revealed that *Candida* spp. were slightly more dominant overall, while *Saccharomyces sensu stricto* was consistently present across all donors (Figure 4B). Notably, Dixon agar selectively enriched for *Candida* and *Saccharomyces*, whereas MSA supported a broader diversity of yeasts, including additional genera categorized as “other.” The greater representation of non-*Candida* yeasts in MSA, particularly from Donors 1 and 3, indicates that this medium is more permissive for recovering a wider spectrum of gut yeasts. Donor-specific variation was also evident, with Donor 2 showing particularly high proportions of *Candida* across both media, suggesting strong colonization by this genus.

Given the study’s focus on identifying non-pathogenic yeasts with probiotic potential, all *Candida*-positive isolates were excluded from subsequent analyses due to their well-documented association with opportunistic pathogenicity and present significant regulatory and public-acceptance constraints for development as live probiotics.

After this exclusion, a final pool of approximately 95 isolates, consisting of *Saccharomyces* spp. and other non-*Candida* yeasts, was selected for downstream functional characterization. This combination of phenotypic and molecular screening established a robust framework for distinguishing commensal yeasts from those with pathogenic traits, thereby enhancing the reliability of candidate selection for probiotic evaluation.

### 3.3. Probiotic Characterization

Following molecular screening, a total of 95 non-*Candida* isolates were retained for probiotic evaluation. To ensure biosafety, 95 yeast isolates were subjected to hemolytic, DNase, and coagulase assays. Among these, 29 displayed hemolytic activity, 21 exhibited DNase activity, and 67 showed coagulase activity. Venn diagram analysis revealed overlapping traits, with 16 isolates positive for all three virulence markers (Figure 5). Based on these results, 69 isolates were excluded, and 26 strains that lacked detectable virulence-associated activities were considered safe and selected for further characterization. The list of yeast isolates showing positive and negative results is provided in Appendix A.

The gastrointestinal resilience of the 26 safe isolates was then assessed under simulated stress conditions. Growth in YPD broth adjusted to pH 2.0 and YPD supplemented with 0.5% *w*/*v* bile salts was compared to standard YPD controls (Figure 6). Several isolates demonstrated the capacity to withstand either acidic pH or bile exposure, indicating strain-specific physiological adaptations. Notably, 20 isolates maintained ≥50% survival in at least one condition, highlighting their potential to persist under major gastrointestinal stresses. Although dual tolerance was less common, the ability of individual strains to adapt to either acid or bile stress supports their candidacy as probiotic agents.

Functional traits of the 26 isolates were further investigated, focusing on antimicrobial activity and bile salt hydrolase (BSH) production. Antimicrobial assays against *Staphylococcus aureus* ATCC 6538P, *Escherichia coli* ATCC 8739, *E. coli* O157:H7, *Salmonella typhimurium* DMST 48437, and *Salmonella enteritidis* S003 revealed considerable variation in inhibition profiles (Table 4). A subset of isolates, including Y6, Y22, Y42, Y48, Y55, Y56, Y73, Y105, and Y127, exhibited broad-spectrum inhibitory activity against both Gram-positive and Gram-negative pathogens. Other strains demonstrated narrower inhibition, restricted to specific bacterial targets, while a minority showed negligible antimicrobial effects.

Assessment of BSH activity demonstrated that all 26 isolates were positive for intracellular BSH production, indicating a conserved capacity for bile salt deconjugation. However, no extracellular BSH activity was detected on TDCA-supplemented agar, suggesting that BSH enzymes remain cell-associated. This implies that any cholesterol-lowering or bile metabolism-modulating effects would depend on yeast survival and metabolic activity within the gastrointestinal tract.

## 4. Discussion

The successful isolation of probiotic yeasts relies heavily on selecting the right culture media and growth conditions. In this study, the combined use of Dixon and MSA media with carefully adjusted antibiotic levels was effective for recovering yeast from fecal samples while limiting bacterial contamination. Antibiotic supplementation was particularly important for suppressing Gram-negative bacteria, which dominate the gut microbiota, but it did not interfere with yeast growth. Moderate doses were the most suitable, as they avoided the inhibitory effects observed at higher concentrations, such as those affecting *Rhodotorula mucilaginosa*. Growth curve analysis further confirmed that low and moderate antibiotic levels did not impair yeast viability, supporting their use in medium design. These results show that antibiotic supplementation reduces bacterial growth and changes the gut microbiota community, thereby creating a more favorable environment for yeast recovery from complex microbial samples [34,35].

Temperature was also an important factor shaping yeast recovery. Yeast counts were consistently higher at 37 °C than at 30 °C, indicating that fecal yeasts are physiologically adapted to the human gut environment. This agrees with previous reports that gastrointestinal yeasts, particularly *Saccharomyces* spp., grow best under conditions that mimic host body temperature [36,37]. The consistent advantage of 37 °C across both media highlights the importance of replicating host-like conditions during the screening process. Such adaptation may also increase the likelihood that these isolates survive and colonize under in vivo conditions, strengthening their potential as probiotic candidates.

In addition to temperature, oxygen availability further distinguished growth responses. Although yeasts are generally considered facultative anaerobes, the isolates obtained from gut samples proliferated more robustly under anaerobic conditions than under aerobic ones [38]. This finding suggests that gut-derived yeasts may have enhanced tolerance to oxygen limitation, a trait advantageous for persistence and activity in the gastrointestinal tract [39]. Interindividual variation among volunteers was explaining only a small proportion of total variance. This likely only reflects natural differences in gut yeast abundance driven by diet, lifestyle, and host physiology. Among the tested conditions, MSA with a moderate antibiotic dose, incubated at 37 °C under anaerobic conditions, emerged as the most effective formulation, balancing bacterial suppression with yeast diversity while closely mimicking host-relevant physiology.

These data demonstrate that systematic optimization of medium composition, antibiotic pressure, and host-like culture conditions substantially improves yeast recovery from complex fecal samples. Although previous studies have isolated the gut fungi, few have directly evaluated medium parameters in an integrated manner, and none have tested the exact combination used here [40,41]. Our findings therefore refine existing approaches by identifying a reproducible set of conditions that maximizes yeast recovery while maintaining compatibility with downstream probiotic screening.

The choice of medium also shaped the diversity of yeast genera recovered. While Dixon medium primarily supported the growth of *Saccharomyces sensu stricto* and *Candida* spp., MSA allowed for the recovery of a broader range of genera, suggesting that its nutrient profile is more permissive for diverse gut-adapted yeasts. Despite this diversity, *Candida* remained the dominant genus in most samples, particularly from certain donors. Given its well-established association with opportunistic pathogenicity this creating significant regulatory, safety, and public-acceptance challenges for probiotic use. Moreover, the taxonomic complexity and high strain-level variability within *Candida* would require extensive virulence and genomic assessments beyond the scope of our initial screening. Thus all *Candida*-positive isolates were excluded from further characterization.

This step highlights the need to pair medium optimization with rigorous safety criteria to ensure that only beneficial yeast is advanced as probiotic candidates. After exclusion, 127 non-*Candida* isolates, mainly belongs to *Saccharomyces* and other safe genera, were retained for functional assessment. Safety evaluation represents another critical stage in probiotic selection. Although yeasts such as *Saccharomyces cerevisiae* are widely regarded as Generally Recognized as Safe (GRAS), mounting evidence indicates that even GRAS strains can display opportunistic traits under certain conditions. Conversely, taxa traditionally considered opportunistic does not always exert adverse effects and may, in some contexts, provide probiotic benefits [42,43]. These examples underscore that probiotic assessment must occur at the strain level, not simply by genus or species classification. In this study, all isolates were screened for hemolytic, DNase, and coagulase activities. Although coagulase testing is not mechanistically relevant to fungal pathogenicity, as it originates from bacterial virulence assays; it was incorporated to maintain methodological continuity with earlier probiotic screening frameworks. The finding that 69 isolates displayed one or more of these traits emphasizes the variability in pathogenic potential within yeast populations and reinforces the necessity of stringent safety testing. Ultimately, only 26 isolates were retained as safe candidates, representing a relatively small fraction of the original collection. This outcome demonstrates that medium formulation alone cannot guarantee the recovery of safe yeast strains, and that functional safety evaluation is indispensable for selecting true probiotic candidates.

The functional characterization of the safe isolates revealed several promising probiotic traits. Antimicrobial screening against enteric pathogens, including *Staphylococcus aureus*, *Escherichia coli* (commensal and pathogenic O157:H7), *Salmonella typhimurium*, and *Salmonella enteritidis*, showed considerable variability across strains. A subset, including Y6, Y22, Y42, Y48, Y55, Y56, Y73, Y105, and Y127, displayed broad-spectrum inhibitory activity, suggesting the production of antimicrobial metabolites relevant to gut health. These observations are consistent with previous reports describing yeast-derived killer toxins, organic acids, and volatile metabolites with antibacterial effects [44,45]. Notably, inhibition was observed against both Gram-positive and Gram-negative species, enhancing the potential utility of these strains as probiotics across diverse host environments. In contrast, other isolates exhibited weaker or narrower inhibition profiles, or none, underscoring the strain-specific nature of antimicrobial metabolite production in yeasts. Nevertheless, further studies are needed to determine whether these isolates exert post-antibiotic effects or influence bacterial biofilm formation, as such insights would contribute to a more comprehensive understanding of their ecological roles and therapeutic potential as gut-derived yeasts.

Bile salt hydrolase (BSH) activity provided further evidence of probiotic potential. All isolates demonstrated positive intracellular BSH activity, confirming their ability to hydrolyze conjugated bile salts upon cell disruption. This activity is closely linked to cholesterol-lowering effects, as bile salt deconjugation reduces reabsorption and promotes cholesterol excretion. The universal presence of intracellular activity suggests that BSH is a conserved functional trait among gut yeasts. By contrast, extracellular BSH activity was absent, indicating that the enzyme remains cell-associated rather than secreted. Functionally, this implies that cholesterol-lowering capacity depends on the survival and metabolic activity of live yeast cells in the gastrointestinal tract. Although the present study assessed only the presence or absence of activity, individual strains may differ in the magnitude of their intracellular BSH function, potentially contributing to variation in physiological impact. Further quantitative assessment of BSH activity would allow clearer differentiation among isolates and could clarify how these differences relate to cholesterol metabolism or broader probiotic functionality.

Altogether, this work demonstrates the feasibility of selectively isolating gut-associated yeasts using carefully optimized media and culture conditions, followed by rigorous safety and functional evaluation. By integrating antibiotic modulation, host-like growth conditions, and strain-level screening, we established a workflow that not only minimizes bacterial contamination but also enriches yeasts with desirable probiotic properties. Although only a small fraction of the initial isolates met the stringent safety criteria, the retained strains displayed relevant functional traits, including antimicrobial activity and bile salt hydrolase production, underscoring their potential as probiotic candidates. These findings highlight the importance of combining medium formulation with comprehensive characterization to identify safe and effective yeast probiotics. Future studies should focus on in vivo validation and genome-level profiling of these strains to confirm their safety, mechanisms of action, and long-term contributions to host health.

## 5. Conclusions

This study established a practical and reproducible workflow for isolating and characterizing gut-associated yeasts, integrating optimized medium formulation, host-relevant culture conditions, and rigorous safety and functional screening. By combining Dixon and MSA media with moderate antibiotic supplementation, incubation at 37 °C, and anaerobic growth, we successfully minimized bacterial interference while recovering diverse yeast populations, including *Saccharomyces* and other non-*Candida* genera. Although only a fraction of the initial isolates passed stringent safety evaluations, several strains, including Y6, Y22, Y42, Y48, Y55, Y56, Y73, Y105, and Y127 exhibited broad antimicrobial activity against both Gram-positive and Gram-negative enteric pathogens. Among these, isolates Y44 and Y55 further demonstrated robust bile and acid tolerance, underscoring their potential for gastrointestinal survival. The retained yeast candidates also displayed intracellular bile salt hydrolase activity associated with cholesterol-lowering effects. These findings highlight the importance of selective medium design coupled with comprehensive screening for advancing safe and effective yeast-based probiotics. Future studies should validate these candidates in vivo and explore their genetic and mechanism to fully realize their potential in gut health applications.

## Figures and Tables

**Figure 1 jof-11-00885-f001:**
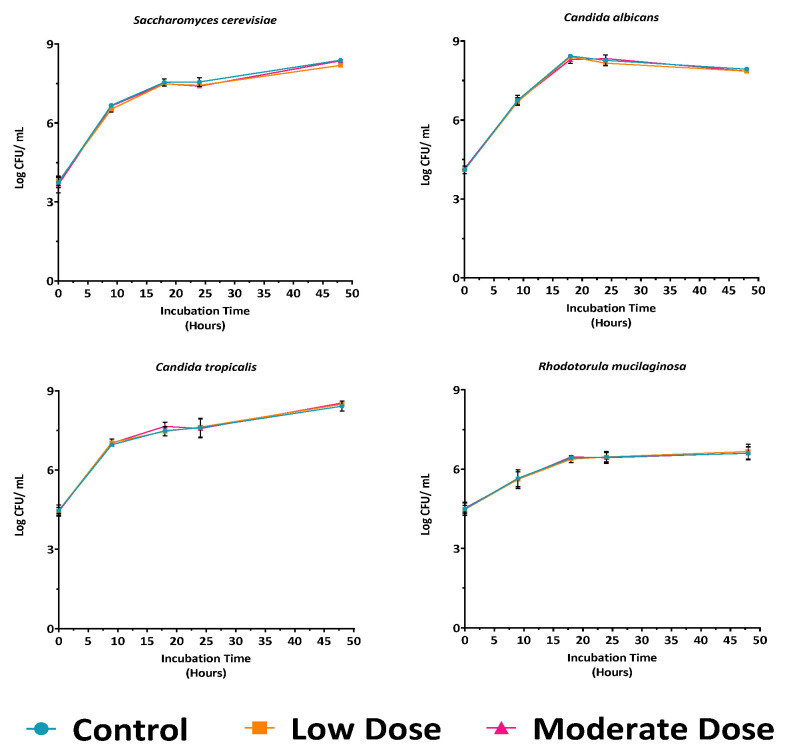
Growth curves of four representative yeast strains in YPD medium under control, low, and moderate antibiotic doses, showing no significant differences in growth patterns at all time points compared to control (*p* > 0.05).

**Figure 2 jof-11-00885-f002:**
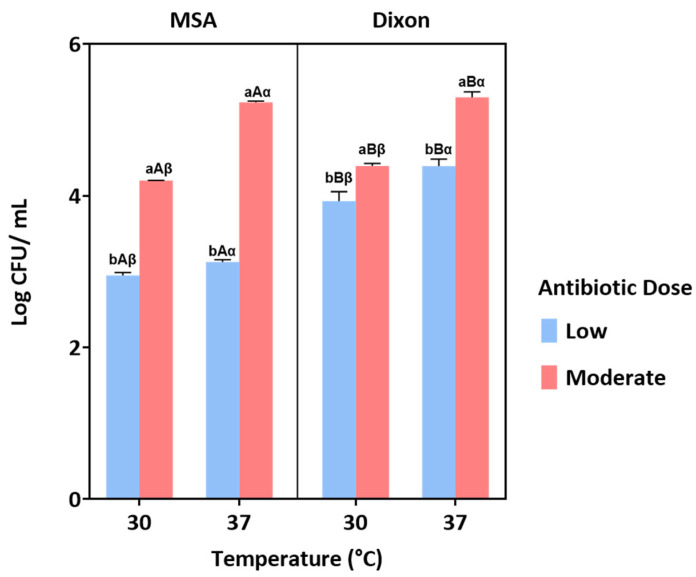
Yeast growth under different conditions of medium (MSA and Dixon), temperature (30 °C and 37 °C), and antibiotic dose (low and moderate). Bars represent the mean ± standard error. Different letters indicate statistically significant differences (*p* < 0.05) based on post hoc analysis. Specifically, ‘a’ and ‘b’ denote significant differences between antibiotic doses, ‘A’ and ‘B’ indicate differences between media, and ‘α’ and ‘β’ represent differences between temperatures.

**Figure 3 jof-11-00885-f003:**
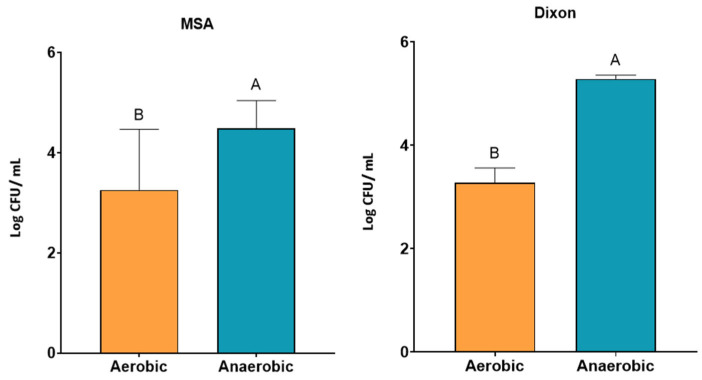
Growth of yeast isolates under aerobic and anaerobic conditions in MSA and Dixon media. Yeast counts (log CFU/mL) were significantly higher under anaerobic conditions compared to aerobic conditions across both media (*p* < 0.05). Bars represent mean ± SD. Different letters (A, B) indicate significant differences between conditions within each medium (two-way ANOVA, *p* < 0.05).

**Figure 4 jof-11-00885-f004:**
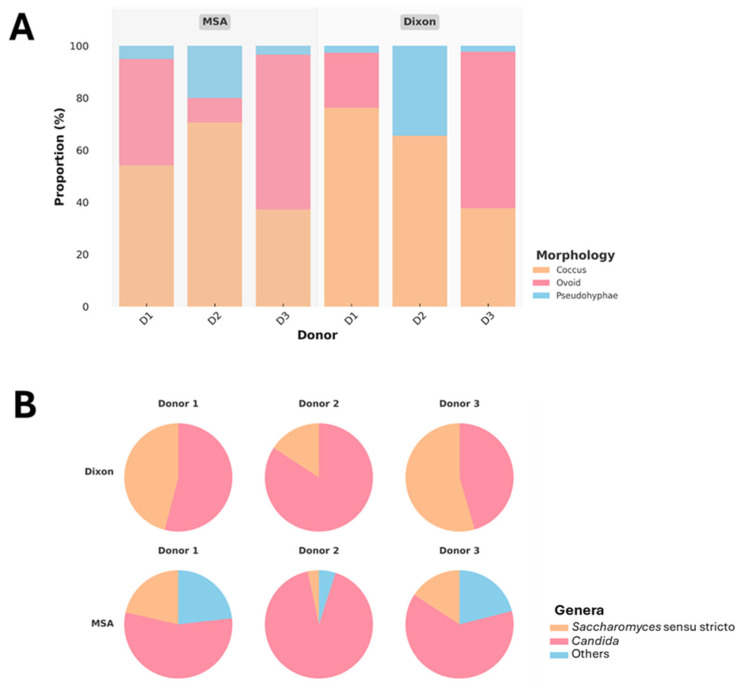
Proportional distribution of yeast morphologies across donors (D1–D3) and media (MSA, Dixon) (**A**), and genus-level classification of isolates based on PCR genotyping (**B**).

**Figure 5 jof-11-00885-f005:**
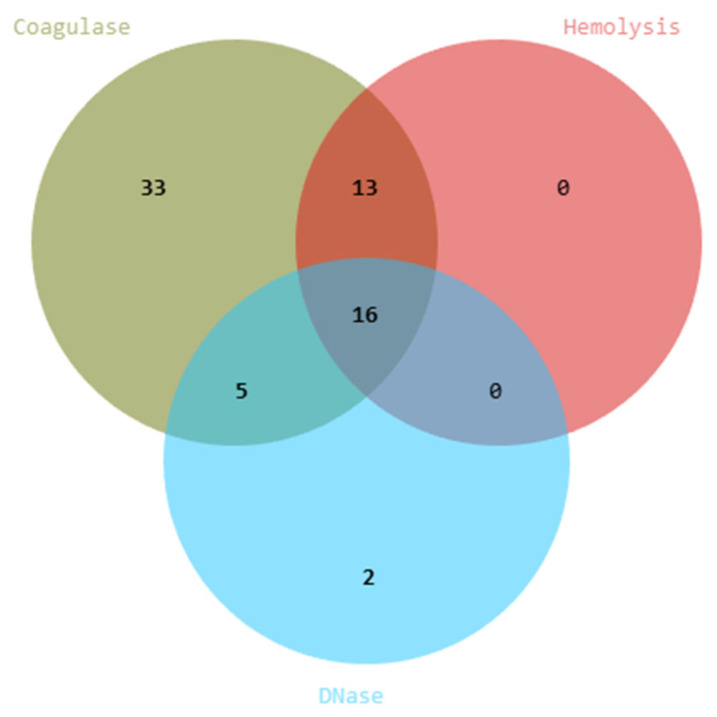
Safety screening and virulence capability of yeast isolates. Venn diagram showing the distribution of isolates positive for hemolytic, DNase, and coagulase activities.

**Figure 6 jof-11-00885-f006:**
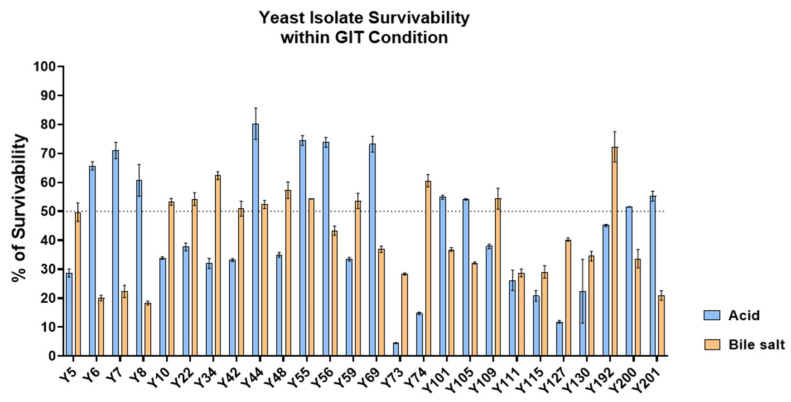
Acid and bile salt tolerance of yeast isolates. Survivability (%) of 26 yeast isolates was evaluated after exposure to acidic conditions (pH 2.0) and 0.5% bile salts. Data represent mean ± SD from triplicate experiments. The dotted line indicates 50% survival, used as the threshold for moderate tolerance.

**Table 1 jof-11-00885-t001:** Mixture of antibiotics and the concentration for yeast isolation.

Antibiotic	Concentration (µg/mL)		
	A (Low)	B (Moderate)	C (High)
Colistin	1	2	4
Chloramphenicol	8	16	32
Vancomcyin	4	8	16

**Table 2 jof-11-00885-t002:** Antibiotic susceptibility of different tested strains at varying antibiotic doses.

Tested Strain	Dose		
	Low (cm)	Moderate (cm)	High (cm)
*Escherichia coli* ATCC 8739	3.17 ± 0.15	4.13 ± 0.12	4.70 ± 0.10
*Salmonella enteritidis* S003	3.23 ± 0.15	4.03 ± 0.12	4.67 ± 0.06
*Staphylococcus aureus* ATCC 6538P	2.10 ± 0.10	3.00 ± 0.10	3.50 ± 0.26
*Rhodotorula mucilaginosa* TBRC-4420	ND	ND	1.37 ± 0.12
*Candida albicans* PCJ94-2	ND	ND	ND
*Candida tropicalis* PYJ100-2	ND	ND	ND
*Saccharomyces cerevisiae* YC1	ND	ND	ND

Data are presented as mean ± standard deviation. ND: No clear zone detected.

**Table 3 jof-11-00885-t003:** Three-way ANOVA representing the factor with significant factors.

Source of Variation	% Var	SS	DF	MS	F	*p*-Value
Temperature	16.0800	2.4810	1	2.4810	598.000	<0.0001
Medium Type	15.2100	2.3470	1	2.3470	624.1000	<0.0001
Antibiotic Dose	54.1500	8.3530	1	8.3530	1899.000	<0.0001
Temperature × Medium Type	0.0590	0.0090	1	0.0090	2.4360	0.1936
Temperature × Antibiotic Dose	4.1060	0.6330	1	0.6330	144.0000	0.0003
Medium Type × Antibiotic Dose	9.5530	1.4740	1	1.4740	345.8000	<0.0001
Temperature × Medium Type × Antibiotic Dose	0.4100	0.0630	1	0.0630	14.8500	0.0182
Volunteers	0.1080	0.0170	4	0.0040		
Volunteers × Medium Type	0.0980	0.0150	4	0.0040		
Volunteers × Antibiotic Dose	0.0110	0.0180	4	0.0040		
Residual		0.0170	4	0.0040		

**Table 4 jof-11-00885-t004:** Antimicrobial activity of yeast isolates against pathogenic bacteria.

Isolate Codes	Inhibition Size Clear Zone (cm)					
	*S. aureus* ATCC 6538P	*E. coli* ATCC 8739	*E. coli* O157:H7	*S. typhimurium* DMST 48437	*S. enteritidis* S003	Spectrum
Y6	1.9 ± 0.0	1.4 ± 0.0	1.1 ± 0.0	1.6 ± 0.0	1.5 ± 0.0	Broad Strong
Y22	1.7 ± 0.0	1.4 ± 0.0	1.2 ± 0.0	1.3 ± 0.0	1.2 ± 0.1	Broad Strong
Y42	1.6 ± 0.1	1.6 ± 0.1	1.8 ± 0.0	1.2 ± 0.1	1.9 ± 0.1	Broad Strong
Y48	1.9 ± 0.0	0.0 ± 0.0	1.4 ± 0.1	1.2 ± 0.0	1.8 ± 0.1	Broad Strong
Y55	1.4 ± 0.0	1.8 ± 0.0	0.0 ± 0.0	1.4 ± 0.1	0.5 ± 0.0	Broad Strong
Y56	1.1 ± 0.1	2.0 ± 0.0	0.9 ± 0.0	1.1 ± 0.1	1.2 ± 0.0	Broad Strong
Y73	1.4 ± 0.1	0.0 ± 0.0	0.0 ± 0.0	1.9 ± 0.1	1.3 ± 0.1	Broad Strong
Y105	1.6 ± 0.0	1.4 ± 0.0	1.8 ± 0.0	0.8 ± 0.0	0.9 ± 0.1	Broad Strong
Y127	1.7 ± 0.0	1.2 ± 0.1	1.5 ± 0.0	0.8 ± 0.0	1.9 ± 0.0	Broad Strong
Y44	0.3 ± 0.0	0.3 ± 0.0	0.0 ± 0.0	0.0 ± 0.0	0.5 ± 0.0	Broad Weak
Y59	0.4 ± 0.0	0.5 ± 0.1	0.0 ± 0.0	0.0 ± 0.0	0.0 ± 0.0	Broad Weak
Y7	0.2 ± 0.1	0.6 ± 0.0	0.4 ± 0.0	0.5 ± 0.0	0.3 ± 0.0	Broad Weak
Y8	0.2 ± 0.0	1.9 ± 0.0	0.6 ± 0.0	0.9 ± 0.0	0.0 ± 0.0	Broad Weak
Y34	0.5 ± 0.0	0.6 ± 0.1	0.8 ± 0.0	0.7 ± 0.1	0.0 ± 0.0	Broad Weak
Y69	0.0 ± 0.0	0.6 ± 0.1	0.0 ± 0.0	1.1 ± 0.0	1.4 ± 0.0	Narrow
Y74	0.0 ± 0.0	1.9 ± 0.1	0.6 ± 0.0	1.7 ± 0.0	1.6 ± 0.0	Narrow
Y101	0.0 ± 0.0	1.5 ± 0.0	0.4 ± 0.1	1.1 ± 0.1	1.2 ± 0.1	Narrow
Y109	0.0 ± 0.0	0.0 ± 0.0	0.2 ± 0.1	0.8 ± 0.0	0.0 ± 0.0	Narrow
Y111	0.0 ± 0.0	0.0 ± 0.0	0.0 ± 0.0	0.0 ± 0.0	0.0 ± 0.0	Narrow
Y115	0.0 ± 0.0	0.0 ± 0.0	0.0 ± 0.0	0.3 ± 0.2	1.3 ± 0.0	Narrow
Y130	0.0 ± 0.0	0.0 ± 0.0	0.0 ± 0.0	0.0 ± 0.0	0.0 ± 0.0	Narrow
Y192	0.0 ± 0.0	1.4 ± 0.1	0.2 ± 0.0	1.9 ± 0.0	0.0 ± 0.0	Narrow
Y200	0.0 ± 0.0	1.4 ± 0.0	1.6 ± 0.0	1.6 ± 0.1	0.4 ± 0.1	Narrow
Y201	0.0 ± 0.0	0.8 ± 0.0	0.5 ± 0.0	0.9 ± 0.0	1.2 ± 0.0	Narrow
Y5	0.0 ± 0.0	0.0 ± 0.0	0.0 ± 0.0	0.3 ± 0.0	0.3 ± 0.0	Narrow
Y10	0.0 ± 0.0	0.8 ± 0.0	0.7 ± 0.0	0.4 ± 0.0	1.6 ± 0.1	Narrow

Broad: inhibits both Gram-positive and Gram-negative bacteria; Narrow: inhibits one Gram group only. Strong: inhibition zone > 1 cm; Weak: <1 cm.

## Data Availability

The original contributions presented in this study are included in the article/Appendix A. Further inquiries can be directed to the first or corresponding author.

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
