# Peer review of "Optimized Selective Media Enhance the Isolation and Characterization of Gut-Derived Probiotic Yeasts"

_jof, 2025, doi:10.3390/jof11120885_

Round 1
Reviewer 1 Report
The study presents an interesting and relevant idea: optimizing media and conditions for isolating gut-associated yeasts and identifying potential probiotic candidates. However, there is an inappropriate selection of certain safety assays that should be properly justified with references.
The manuscript uses coagulase and hemolysis assays as part of the safety screening. While hemolysis is relevant, coagulase is a classical bacterial virulence marker and does not apply to yeasts. Its use here is not biologically justified, and the interpretation of any “positive” results is unclear.
Recommended corrective actions:
Replace or complement these tests with validated fungal pathogenicity markers, such as phospholipase and protease activity, biofilm formation, antifungal susceptibility testing, and genomic screening for known virulence genes.
Clarify that coagulase assays do not measure yeast–bacteria interactions and are not appropriate indicators of yeast safety.
Minor Comments:
The term “pseudohyphal” is used as a safety exclusion criterion, but pseudohyphae are common in many non-pathogenic yeasts. Please provide justification and appropriate references.
Add details on medium composition, incubation conditions, and times for reproducibility.
Author Response
Reviewer 1
Comment 1: The study presents an interesting and relevant idea: optimizing media and conditions for isolating gut-associated yeasts and identifying potential probiotic candidates. However, there is an inappropriate selection of certain safety assays that should be properly justified with references.
The manuscript uses coagulase and hemolysis assays as part of the safety screening. While hemolysis is relevant, coagulase is a classical bacterial virulence marker and does not apply to yeasts. Its use here is not biologically justified, and the interpretation of any “positive” results is unclear.
Recommended corrective actions: Replace or complement these tests with validated fungal pathogenicity markers, such as phospholipase and protease activity, biofilm formation, antifungal susceptibility testing, and genomic screening for known virulence genes. Clarify that coagulase assays do not measure yeast–bacteria interactions and are not appropriate indicators of yeast safety.
Answer:
We thank the reviewer for this valuable comment and fully acknowledge that the classical coagulase assay is a bacterial virulence marker and not a standard pathogenicity indicator for yeasts. Its biological relevance for fungal safety assessment is indeed limited, and we agree that interpretation of ‘positive’ results in yeasts is difficult.
To better accommodate this issue, we have omitted coagulase from safety assessment and add it for additional testing at section 2.6.2 and add this sentence “Coagulase activity was included as an additional precautionary measure to screen for any unexpected virulence-associated traits”. (Line 256-257). We also add sentence to the manuscript (Line 565-567) “Although coagulase testing is not mechanistically relevant to fungal pathogenicity, as it originates from bacterial virulence assays; it was incorporated to maintain methodological continuity with earlier probiotic screening frameworks.”
Comment 2 The term “pseudohyphal” is used as a safety exclusion criterion, but pseudohyphae are common in many non-pathogenic yeasts. Please provide justification and appropriate references. Add details on medium composition, incubation conditions, and times for reproducibility.
Answer:
We thank the reviewer for this insightful comment. Pseudohyphal formation may reflect increased adhesive or invasive potential. We use these methods as an initial screening step to exclude strains showing extensive filamentation under standard laboratory conditions, which can be associated with higher invasion potential in certain opportunistic yeasts (e.g., Candida spp.). We have added the following clarification in lines 199–202: “Yeasts that did not produce pseudohyphae or hyphae were selected for further characterization. Given that pseudohyphal formation may reflect increased adhesive or invasive potential, these isolates were excluded as a precaution to maintain a conservative safety threshold for probiotic selection”
In addition, following the reviewer’s recommendation, we have now provided detailed information on the media composition, incubation conditions, and incubation times used for pseudohyphal observation and other morphological assessments. These details are now included in the Methods section (line 212-217) to improve reproducibility and clarity.
Reviewer 2 Report
Section 2.3 / Methodology Description:Recommendation: Briefly discuss in the Introduction or Methods the specific compositional features of Dixon and MSA media (e.g., specific carbon/nitrogen sources or vitamins) and their potential advantages in mimicking the gut environment or supporting fastidious yeast growth.
Section 2.6.2 / Tolerance Assay: Acid and bile salt tolerance were assessed by measuring OD420 for survival rate..Recommendation: Justify the choice of the OD method in the methods.
Section 3.1.2 / Results Presentation: The three-way ANOVA indicates that variation among volunteers was significant (p=0.0182). However, the Discussion does not delve into the implications of this inter-individual variation on the isolation outcomes,Recommendation: Briefly comment in the Discussion on the potential sources of inter-individual variation (e.g., diet, lifestyle) and its implications for defining a "core" probiotic yeast community。
Section 3.3 / Functional Characterization: The study tested antimicrobial activity but did not explore whether these yeast isolates might produce post-antibiotic effects or inhibit bacterial biofilm formation, which are important additional functional attributes for probiotics.Recommendation: Mention these aspects as future research directions in the Discussion。
Section 5 / Conclusion: The conclusion mentions providing insights for "sorghum bran utilization," but the direct application of this study is in probiotic development and gut health. Recommendation: Reframe the conclusion to more directly reflect the main findings of this study and their specific significance in the microbiome/probiotic field.
Section 5 / Conclusion: Recommendation: Highlight a few specific isolate codes in the conclusion that represent the most promising leads for future development based on the cumulative data.pathway)。
The focus of this paper lies in enhancing yeast recovery rates through optimized selection media. The focus of this paper lies in enhancing yeast recovery rates through optimized selection media. It is recommended to highlight the core advantage of this study by comparing it with relevant literature (which did not perform media optimization
Lines118-120 (Table 1) and Lines 334-335: The study employed a specific antibiotic cocktail (colistin, chloramphenicol, vancomycin) at moderate concentrations, primarily citing CLSI guidelines. However, the manuscript does not fully elaborate on the rationale for selecting this particular combination,Briefly justify the strategic choice of this specific antibiotic combination in the Methods or Discussion, explaining how it suppresses predominant gut bacteria while sparing yeasts.
Line 222: To streamline the content of the manuscript, the section heading for Section 2.6 may be retained, while the content of Section 2.6 may be deleted. The text should then transition directly to the corresponding content in Section 2.6.1.
Line 352: Please provide the yeast recovery rates for all medium combinations in the three-factor ANOVA.
Lines 395-405 (Section 3.2):Recommendation:Based on the morphology of pseudohyphae and the positive PCR results, all Candida isolates were excluded. It is suggested that more comprehensive reasons be provided in the discussion section to verify why all Candida isolates were excluded.
Lines 443-444 & Table 4 (Section 3.3): The antimicrobial activity results are presented qualitatively (e.g., "Broad Strong") and with inhibition zone sizes. However, the criteria for categorizing "Strong" vs. "Weak" inhibition are not explicitly defined.
Recommendation: Define the criteria for "Strong," "Weak," and "Narrow" spectrum in the Methods or the legend for Table 4
Lines 477-483: This paragraph may be omitted, as its content is redundant with the following paragraph.
Lines 503-510 (Section 3.3) and Discussion: The study found that all 26 safe isolates possessed intracellular but not extracellular BSH activity.Recommendation: Consider discussing the potential variation in BSH activity levels among strains or note that future work could quantify this to differentiate functional potency.
Lines 520-522: The article indicates in multiple places that MSA can isolate a greater variety of yeast strains compared to Dixon agar. Then why does it conclude that the optimal culture conditions are 37°C anaerobic conditions with a medium antibiotic dose on Dixon agar? Is it because among the final 26 strains selected, more originated from isolates on Dixon agar than from MSA? The article does not explain this.
Author Response
Reviewer 2
Q1: Briefly discuss in the Introduction or Methods the specific compositional features of Dixon and MSA media (e.g., specific carbon/nitrogen sources or vitamins) and their potential advantages in mimicking the gut environment or supporting fastidious yeast growth.
Answer: Thank you for your kind comments we have added to the manuscript at line 178-182.
Q2: Justify the choice of the OD method in the methods.
Answer: Thank you for looking upon this. There is a minor error in the previous manuscript version. The sentence now has now been corrected to the optical density measurement to OD₆₀₀ in this study and we have added the appropriate citation.
Q3: The three-way ANOVA indicates that variation among volunteers was significant (p=0.0182). However, the Discussion does not delve into the implications of this inter-individual variation on the isolation outcomes. Briefly comment in the Discussion on the potential sources of inter-individual variation (e.g., diet, lifestyle) and its implications for defining a "core" probiotic yeast community.
Answer: Thank you for your suggestion. Although the effect of antibiotic dose was not statistically significant (p = 0.0182), we agree that inter-individual variation among volunteers may have contributed to the observed differences. We have now included additional details on the volunteer effect in the Results section (Lines 378–385) and expanded the Discussion to address this point (Lines 531–534).
Q4: The study tested antimicrobial activity but did not explore whether these yeast isolates might produce post-antibiotic effects or inhibit bacterial biofilm formation, which are important additional functional attributes for probiotics. Mention these aspects as future research directions in the Discussion.
Answer: Thank you for your suggestions, we have added new research directions to the discussion part Line 585-589.
Q5: The conclusion mentions providing insights for "sorghum bran utilization," but the direct application of this study is in probiotic development and gut health. Reframe the conclusion to more directly reflect the main findings of this study and their specific significance in the microbiome/probiotic field.
Answer: We thank the reviewer for the comment; however, we would like to clarify that the present study does not include any component related to sorghum bran utilization. This appears to be an inadvertent misunderstanding, as no such material or application is mentioned in the manuscript. Our work is focused solely on the isolation, safety evaluation, and functional characterization of gut-derived yeast isolates with relevance to probiotic development and gut health. The conclusion has been reviewed to ensure that it accurately reflects these objectives and findings.
Q6: Highlight a few specific isolate codes in the conclusion that represent the most promising leads for future development based on the cumulative data pathway.
Answer: Thank you for your suggestion. We have added a new sentence to the Conclusion highlighting the isolates with broad and strong antibacterial activity (Y6, Y22, Y42, Y48, Y55, Y56, Y73, Y105, and Y127), as well as those demonstrating both gastric and bile tolerance (Y44 and Y55).
Q7: The focus of this paper lies in enhancing yeast recovery rates through optimized selection media. The focus of this paper lies in enhancing yeast recovery rates through optimized selection media. It is recommended to highlight the core advantage of this study by comparing it with relevant literature (which did not perform media optimization)
Answer: Thank you for your comment. While we agree that emphasizing the advantages of our optimized media is valuable, a direct comparison with previous studies is not feasible because no existing work has evaluated yeast recovery using the same combination of parameters (medium type, antibiotic dose, temperature, and anaerobic conditions). To address your suggestion, we have added a brief clarification in the discussion Line 537-543 highlighting how our media optimization approach fills this methodological gap without attempting an inappropriate one-to-one comparison.
Q8: Lines118-120 (Table 1) and Lines 334-335: The study employed a specific antibiotic cocktail (colistin, chloramphenicol, vancomycin) at moderate concentrations, primarily citing CLSI guidelines. However, the manuscript does not fully elaborate on the rationale for selecting this combination Briefly justify the strategic choice of this specific antibiotic combination in Methods or Discussion, explaining how it suppresses predominant gut bacteria while sparing yeasts.
Answer: We appreciate the reviewer’s comment. The rationale for the selected antibiotic combination has now been briefly clarified in the manuscript. As noted, each antibiotic was used at concentrations guided by CLSI susceptibility breakpoints for Enterobacteriaceae, a bacterial group that is abundantly represented in the human gut and frequently overgrows on non-selective media during yeast isolation. The combination of colistin, chloramphenicol, and vancomycin was strategically selected because it effectively suppresses a broad range of dominant gut bacteria, including Gram-negative and Gram-positive bacteria. We have added this information to the manuscript Line 117-122
Q9: Line 222, To streamline the content of the manuscript, the section heading for Section 2.6 may be retained, while the content of Section 2.6 may be deleted. The text should then transition directly to the corresponding content in Section 2.6.1.
Answer: Thank you for your suggestion, we have removed the text and move from 2.6 to 2.6.1 directly
Q10: Please provide the yeast recovery rates for all medium combinations in the three-factor ANOVA.
Answer: Thank you for your suggestion, we have added this recovery rates in supplementary table 1
Q11: Lines 395-405 (Section 3.2), Based on the morphology of pseudohyphae and the positive PCR results, all Candida isolates were excluded. It is suggested that more comprehensive reasons be provided in the discussion section to verify why all Candida isolates were excluded.
Answer: Thank you for your suggestion. We have added additional information regarding pseudohyphal formation in Lines 199–202 and clarified the relevant details for Candida in Lines 433–434.
Q12: Lines 443-444 & Table 4 (Section 3.3): The antimicrobial activity results are presented qualitatively (e.g., "Broad Strong") and with inhibition zone sizes. However, the criteria for categorizing "Strong" vs. "Weak" inhibition are not explicitly defined. Define the criteria for "Strong," "Weak," and "Narrow" spectrum in the Methods or the legend for Table 4
Answer: Thank you for your suggestions. We have added the footnotes to table 4 to enhance clarity.
Q14: Lines 477-483 This paragraph may be omitted, as its content is redundant with the following paragraph.
Answer: Thank you for your suggestions. We have omitted the sentence on respective line
Q15: Lines 503-510 (Section 3.3) and Discussion The study found that all 26 safe isolates possessed intracellular but not extracellular BSH activity. Consider discussing the potential variation in BSH activity levels among strains or note that future work could quantify this to differentiate functional potency.
Answer: Thank you for your suggestion. We have added more information in discussion section Line 598-603.
Q16: Lines 520-522 The article indicates in multiple places that MSA can isolate a greater variety of yeast strains compared to Dixon agar. Then why does it conclude that the optimal culture conditions are 37°C anaerobic conditions with a medium antibiotic dose on Dixon agar? Is it because among the final 26 strains selected, more originated from isolates on Dixon agar than from MSA? The article does not explain this.
Answer: Thank you for your suggestion. We acknowledge that the original text incorrectly referred to Dixon agar as yielding the optimal culture conditions. This has now been corrected to reflect that MSA, not Dixon Agar, provided the highest diversity of isolates under 37 °C, anaerobic conditions with a medium antibiotic concentration. We have revised the relevant sentences and added clarification in the Discussion (Line 534)
Reviewer 3 Report
Dear authors:
The authors of this study tried to screen yeasts with probiotic potentials by optimizing selective medium and characterizing the yeasts isolated from fecal samples. Some suggestions/comments and questions are raised as following:
- Among the 305 isolates recovered in this study, 26 strains may serve as probiotic candidate by safety screening and functional assays. The scientific name and isolation informations of these 26 yeasts are suggested listed.
- Among the 305 isolates recovered in this study, PCR positive Candida species and isolates with pseudohyphal morphology were excluded for evaluation as probiotic candidates. Some questions are raised to the practice:
- What are the methods and culture conditions used to observe the pseudohyphae production? by direction observation on MSA/ Dixon medium or by Dalmau plates?
- What are the reasons to exclude these yeasts with pseudohyphae morphology. As we know, some yeasts can produce pseudohyphae/ hyphae, but they can be regarded as GRAS species in some country, ex. Yarrowia lipolytica, Saccharomycopsis fibuligera.
- What are the reasons to exclude Candida species. As we know, some anamorphic yeasts in ascomycete were named as Candida species before 2008, because these yeasts lacking differentiated sexual or genus leveled morphological characteristics. So Candida species demonstrated polyphyletic, and showed distinctly difference in morphology and physiological characteristics each other. Although some Candida species have been reclassified or moved to other taxa, but some non-pathogenic Candida species retained in the genus. What are the reasons to exclude these retained Candida species. At other side, some pathogenic Candida species, Candida glabrata were reclassified in other genus as Nakaseomyces glabratus. Did the pathogenic yeasts were excluded?
- Two functional assays were performed for screening of candidate probiotic in this study, antimicrobial activity and enzyme activity of BSH. Any other important functional assay were performed in this study?
- In the same data set, the numbers of significant figures after the decimal point of each measurement or statistical analysis are suggested to be the same, ex. Table 2, Table 3, Table 4.
- 1. Please check the indicator of Y-axis, CFU/mL, are they correct?
- L427-430. “To ensure biosafety, 95 yeast isolates were subjected to hemolytic, DNase, and coagulase assays. Among these, 34 displayed hemolytic activity, 19 exhibited DNase activity, and 69 showed coagulase activity. ---- virulence markers (Figure 5)” and L530_ “The finding that 69 isolates displayed one 530 or more of these traits emphasizes----”. The numbers of isolates described in these sentence is inconsistent with those listed on 5. Please check that.
Dear authors:
The authors of this study tried to screen yeasts with probiotic potentials by optimizing selective medium and characterizing the yeasts isolated from fecal samples. Some suggestions/comments and questions are raised as following:
- Among the 305 isolates recovered in this study, 26 strains may serve as probiotic candidate by safety screening and functional assays. The scientific name and isolation informations of these 26 yeasts are suggested listed.
- Among the 305 isolates recovered in this study, PCR positive Candida species and isolates with pseudohyphal morphology were excluded for evaluation as probiotic candidates. Some questions are raised to the practice:
- What are the methods and culture conditions used to observe the pseudohyphae production? by direction observation on MSA/ Dixon medium or by Dalmau plates?
- What are the reasons to exclude these yeasts with pseudohyphae morphology. As we know, some yeasts can produce pseudohyphae/ hyphae, but they can be regarded as GRAS species in some country, ex. Yarrowia lipolytica, Saccharomycopsis fibuligera.
- What are the reasons to exclude Candida species. As we know, some anamorphic yeasts in ascomycete were named as Candida species before 2008, because these yeasts lacking differentiated sexual or genus leveled morphological characteristics. So Candida species demonstrated polyphyletic, and showed distinctly difference in morphology and physiological characteristics each other. Although some Candida species have been reclassified or moved to other taxa, but some non-pathogenic Candida species retained in the genus. What are the reasons to exclude these retained Candida species. At other side, some pathogenic Candida species, Candida glabrata were reclassified in other genus as Nakaseomyces glabratus. Did the pathogenic yeasts were excluded?
- Two functional assays were performed for screening of candidate probiotic in this study, antimicrobial activity and enzyme activity of BSH. Any other important functional assay were performed in this study?
- In the same data set, the numbers of significant figures after the decimal point of each measurement or statistical analysis are suggested to be the same, ex. Table 2, Table 3, Table 4.
- 1. Please check the indicator of Y-axis, CFU/mL, are they correct?
- L427-430. “To ensure biosafety, 95 yeast isolates were subjected to hemolytic, DNase, and coagulase assays. Among these, 34 displayed hemolytic activity, 19 exhibited DNase activity, and 69 showed coagulase activity. ---- virulence markers (Figure 5)” and L530_ “The finding that 69 isolates displayed one 530 or more of these traits emphasizes----”. The numbers of isolates described in these sentence is inconsistent with those listed on 5. Please check that.
Author Response
Reviewer 3
Comments
The authors of this study tried to screen yeasts with probiotic potentials by optimizing selective medium and characterizing the yeasts isolated from fecal samples. Some suggestions/comments and questions are raised as following.
Q1: Among the 305 isolates recovered in this study, 26 strains may serve as probiotic candidate by safety screening and functional assays. The scientific name and isolation informations of these 26 yeasts are suggested listed.
Answer: Thank you for your suggestion. At this stage, we only have confirmed information on whether the isolates belong to Saccharomyces or non-Candida, non-Saccharomyces yeasts; therefore, we have retained strain codes in the manuscript to avoid providing incomplete or potentially inaccurate species-level names. We appreciate your recommendation, and we plan to present the full scientific names and detailed isolation information in future work once all identifications are fully verified.
Q2 What are the methods and culture conditions used to observe the pseudohyphae production? by direction observation on MSA/ Dixon medium or by Dalmau plates?
Answer: Thank you for your helpful comment. In our study, pseudohyphal formation was assessed by direct microscopic observation on half-strength YPD agar, which we selected as a nutrient-limited condition commonly used to induce filamentous growth in yeasts. We have now added these methodological details to the revised Methods section for clarity (Line 213-215). We appreciate your suggestion regarding the use of Dalmau plates and thank you for highlighting the importance of specifying culture conditions.
Q3 What are the reasons to exclude these yeasts with pseudohyphae morphology. As we know, some yeasts can produce pseudohyphae/ hyphae, but they can be regarded as GRAS species in some country, ex. Yarrowia lipolytica, Saccharomycopsis fibuligera.
Answer: Thank you for the comment. We excluded pseudohyphae-forming yeasts because our goal is to identify strains suitable for direct human probiotic use, where safety standards are stricter than for food fermenters. While some species that form pseudohyphae (e.g., Yarrowia lipolytica, Saccharomycopsis fibuligera) are considered GRAS, this status usually applies to their use as fermentation or processing organisms that are removed or inactivated before consumption. This does not guarantee safety when the cells are consumed alive as probiotics. In response to the reviewer’s suggestion, we have added the following clarification in lines 215–218: “Yeasts that did not produce pseudohyphae or hyphae were selected for further characterization. Given that pseudohyphal formation may reflect increased adhesive or invasive potential, these isolates were excluded as a precaution to maintain a conservative safety threshold for probiotic selection”. We hope this addition is able to clearly communicate the rationale and purpose of this exclusion criteria
Q4 What are the reasons to exclude Candida species. As we know, some anamorphic yeasts in ascomycete were named as Candiis ableblepecies before 2008, because these yeasts lacking differentiated sexual or genus leveled morphological characteristics. So Candida species demonstrated polyphyletic and showed distinct differences in morphology and physiological characteristics each other. Although some Candida species have been reclassified or moved to other taxa, but some non-pathogenic Candida species retained in the genus. What are the reasons to exclude these retained Candida species. At other side, some pathogenic Candida species, Candida glabrata were reclassified in other genus as Nakaseomyces glabratus. Did the pathogenic yeasts were excluded?
Answer: Thank you for your insightful comment. We excluded Candida species based on both safety considerations and practical relevance to our study, which aims to identify strains suitable for potential human probiotic development. Although Candida is a polyphyletic genus and some species are non-pathogenic, the genus remains strongly associated with opportunistic infections, creating significant regulatory, safety, and public-acceptance challenges for probiotic use. Moreover, the taxonomic complexity and high strain-level variability within Candida would require extensive virulence and genomic assessments beyond the scope of our initial screening. To avoid any ambiguity, we removed all isolates identified as Candida through our genomic analysis using species-specific primers, including those that have been historically reclassified from clinically relevant taxa such as Candida glabrata. This precautionary, safety-first approach ensures that only yeast species with clear records of safe human use were included in our study. We also add sentence in our manuscript to acknowledge this at line 433-434.
Q4 Two functional assays were performed for screening of candidate probiotic in this study, antimicrobial activity and enzyme activity of BSH. Any other important functional assay were performed in this study?
Answer: Thank you for your question. In this study, our main objective was to investigate how different media and environmental conditions (antibiotic supplementation, temperature, medium composition, and oxygen levels) influence yeast isolation. In addition to antimicrobial activity and BSH assays, we also assessed probiotic potential through survival under simulated gastric and intestinal conditions, as well as conducting safety screening for each strain. These evaluations together provide a foundation assessment of the isolates’ probiotic suitability.
Q5 In the same data set, the numbers of significant figures after the decimal point of each measurement or statistical analysis are suggested to be the same, ex. Table 2, Table 3, Table 4.
Answer: Thank you very much for your suggestion. We have revised the significant figures in each table to ensure consistency within each dataset (Tables 2, 3, and 4). We appreciate your careful review and constructive feedback.
Q6 1. Please check the indicator of Y-axis, CFU/mL, are they correct?
Answer: Thank you for pointing this out. We have re-checked the Y-axis indicator and corrected Figure 1 from CFU/ mL to Log CFU/mL. We sincerely appreciate your careful observation and thank you for helping us improve the clarity of the figure.
Q7 L427-430. “To ensure biosafety, 95 yeast isolates were subjected to hemolytic, DNase, and coagulase assays. Among these, 34 displayed hemolytic activity, 19 exhibited DNase activity, and 69 showed coagulase activity. ---- virulence markers (Figure 5)” and L530_ “The finding that 69 isolates displayed one 530 or more of these traits emphasizes----”. The numbers of isolates described in this sentence is inconsistent with those listed on 5. Please check that.
Answer: Thank you for highlighting this issue. We have carefully rechecked and updated both the text and Figure 5 to ensure consistency. The revised text now states: To ensure biosafety, 95 yeast isolates were subjected to hemolytic, DNase, and coagulase assays. Among these, 29 displayed hemolytic activity, 21 exhibited DNase activity, and 67 showed coagulase activity. Venn diagram analysis revealed overlapping traits, with 16 isolates positive for all three virulence markers (Figure 5).’ To enhance clarity, we have also added a supplementary table summarizing each isolate’s results across all three assays, allowing the overlap to be easily verified. We appreciate the reviewer’s helpful comment and have revised the manuscript accordingly.
Reviewer 4 Report
This study developed an optimized culturomic workflow for the isolation and characterization of intestinal yeasts with probiotic potential. The use of Dixon and Schädler media modified with a moderate mix of antibiotics, incubated anaerobically at 37 °C, reduced bacterial contamination and favored yeast growth. 305 isolates were obtained from samples from three healthy donors, of which, after excluding pseudohyphal forms and Candida species, 127 non-Candida strains remained. Safety screening (hemolysis, DNase, coagulase) selected 26 safe isolates, and gastrointestinal tolerance tests showed that 20 of these survive in a proportion of more than 50% at acidic pH or in the presence of bile salts. Functional evaluation revealed strain-specific antimicrobial activity against pathogens such as Staphylococcus aureus, Escherichia coli (including O157:H7), and Salmonella, with several isolates exhibiting broad-spectrum inhibition. All strains exhibited intracellular bile salt hydrolysis activity, suggesting a possible cell viability-dependent cholesterol-lowering effect. Overall, this practical and reproducible workflow enabled the recovery of safe and functional intestinal yeasts, including Saccharomyces and other non-Candida genera, demonstrating the importance of selective media design and rigorous screening. The results highlight several promising strains as yeast-based probiotics that require in vivo validation and further investigation into mechanisms and potential applications in intestinal health.
I don't have detailed comments.
Author Response
Reviewer 4
Comments: This study developed an optimized culturomic workflow for the isolation and characterization of intestinal yeasts with probiotic potential. The use of Dixon and Schädler media modified with a moderate mix of antibiotics, incubated anaerobically at 37 °C, reduced bacterial contamination and favored yeast growth. 305 isolates were obtained from samples from three healthy donors, of which, after excluding pseudohyphal forms and Candida species, 127 non-Candida strains remained. Safety screening (hemolysis, DNase, coagulase) selected 26 safe isolates, and gastrointestinal tolerance tests showed that 20 of these survive in a proportion of more than 50% at acidic pH or in the presence of bile salts. Functional evaluation revealed strain-specific antimicrobial activity against pathogens such as Staphylococcus aureus, Escherichia coli (including O157:H7), and Salmonella, with several isolates exhibiting broad-spectrum inhibition. All strains exhibited intracellular bile salt hydrolysis activity, suggesting a possible cell viability-dependent cholesterol-lowering effect. Overall, this practical and reproducible workflow enabled the recovery of safe and functional intestinal yeasts, including Saccharomyces and other non-Candida genera, demonstrating the importance of selective media design and rigorous screening. The results highlight several promising strains as yeast-based probiotics that require in vivo validation and further investigation into mechanisms and potential applications in intestinal health.
Answer: We thank the reviewer for the thoughtful and encouraging summary of our work. We agree that in vivo validation and mechanistic investigations will be essential next steps, and these components are planned for future studies to further evaluate the probiotic potential and functional relevance of the identified yeast strains.
Round 2
Reviewer 1 Report
In my view, the authors have adequately addressed the concerns raised and incorporated the necessary revisions. The manuscript has clearly improved, and I consider it suitable for publication in its current form.
n my view, the authors have adequately addressed the concerns raised and incorporated the necessary revisions. The manuscript has clearly improved, and I consider it suitable for publication in its current form.